# Underwater Image Enhancement Based on Histogram-Equalization Approximation Using Physics-Based Dichromatic Modeling

**DOI:** 10.3390/s22062168

**Published:** 2022-03-10

**Authors:** Yan-Tsung Peng, Yen-Rong Chen, Zihao Chen, Jung-Hua Wang, Shih-Chia Huang

**Affiliations:** 1Department of Computer Science, National Chengchi University, Taipei City 116, Taiwan; ytpeng@cs.nccu.edu.tw (Y.-T.P.); 109753204@nccu.edu.tw (Y.-R.C.); 2Department of Mechanical Aerospace Engineering, University of California, San Diego, CA 92093, USA; zic042@ucsd.edu; 3AI Research Center, National Taiwan Ocean University, Keelung City 202, Taiwan; jhwang@ntou.edu.tw; 4Department of Electronic Engineering, National Taipei University of Technology, Taipei City 106, Taiwan

**Keywords:** histogram equalization approximation, physics-based dichromatic model, convex optimization

## Abstract

This work proposes to develop an underwater image enhancement method based on histogram-equalization (HE) approximation using physics-based dichromatic modeling (PDM). Images captured underwater usually suffer from low contrast and color distortions due to light scattering and attenuation. The PDM describes the image formation process, which can be used to restore nature-degraded images, such as underwater images. However, it does not assure that the restored images have good contrast. Thus, we propose approximating the conventional HE based on the PDM to recover the color distortions of underwater images and enhance their contrast through convex optimization. Experimental results demonstrate the proposed method performs favorably against state-of-the-art underwater image restoration approaches.

## 1. Introduction

Underwater exploration has progressed greatly due to underwater remotely operated vehicles (ROVs) mounted with underwater surveillance video cameras or event data recorders [1]. Ones can collect and analyze data through computer vision applications, such as image classification, object detection, semantic segmentation, etc. One of the most critical factors for these systems to work well is that the recorded videos and the captured images are visually clear with good contrast. However, it is not always the case that the visual source has decent visibility [2]. Since the attenuated light reflected by the scene reaches the camera, causing absorption and scattering effects, underwater images often have color distortions and appear hazy and blurry [3].

Visibility improvement in underwater images is always a challenge. Contrast enhancement (CE) is a useful technique to enhance low-contrast images’ visual quality and bring out image details that would be otherwise unseen. Among different approaches for CE, global HE [4] is one of the most widely used methods due to its simple implementation and generally satisfactory results. HE tries to distribute intensity levels (IL) of an image uniformly across the available IL range through a transformation function calculated using the image cumulative distribution function (CDF). It works for general cases; however, the absorbed lights for underwater scenes make the images bluish or greenish, which may cause HE-based methods to under-enhance or over-enhance the images. Therefore, HE may not work well for such images since it does not consider underwater physics characteristics.

On the other hand, several attempts have been made to restore color and sharpness for underwater images using physics-based dichromatic modeling (PDM) [5,6,7,8,9]. Drews et al. [7], inspired by the dark channel prior (DCP) for image dehazing [8], restored underwater images using underwater DCP. Peng et al. [10] adopted image blurriness and light absorption to estimate depth information and background light for underwater scenes for underwater image restoration. Peng et al. [9] generalize the DCP to apply to various nature-degraded scenes, such as underwater, hazy, and sandstorm. However, physics-based dichromatic modeling often can only achieve color restoration, but the restored images may still not have good contrast.

In traditional image enhancement methods, [5,6,11,12,13,14,15,16] have all made related attempts. The scene depths used in [5,12,15] are all obtained from the transmission map (TM), which is derived from the DCP. He et al. [8] proposed to compute the amount of spatially uniform haze by using the darkest channel in the scene to remove haze in natural images. In natural scenes, DCP can be used to estimate TM and scene depth since points closer to the camera have shorter scattering paths. In contrast, points closer to dark scenes are still dark because of the intensity of scattered light they experience is lower. However, these traditional methods may still fail to restore red light for underwater images due to its long wavelengths and low frequencies, which attenuate faster underwater. Therefore, in underwater scenes, RGB channel-based of DCP often only considers the red channel to measure the transmission, resulting in incorrect depth estimation and poor restoration performances. Sequentially, an underwater DCP based only on the blue and green channels was proposed in [6,13,16] to solve the above problems. Additionally, Galdran et al. [14] proposed an approach based on DCP of green, blue, and inverted red channels, called the Red Channel method. Furthermore, Carlevaris-Bianco et al. [11] replaced DCP with the maximum intensity prior for estimating TM, which calculates the difference between the maximum intensity of the red channel and the maximum intensity of the blue and green. However, there are many exceptions to these priors due to the light scattering, absorption, and different lighting conditions in underwater images, which lead to no way for these methods to achieve satisfactory performance.

In recent years, deep learning has made significant progress, especially in the field of computer vision and low-level image processing tasks [17,18,19,20,21,22]. As a pioneering work, Cao et al. [17] proposed to synthesize degraded/clean underwater image pairs based on the image formation model to train a model to estimate scene depth and background light for an underwater image. Li et al. [23] extended it by using a Generative Adversarial Network (GAN) to make restored underwater images more realistic. In addition, Guo et al. [24] employed a residual multi-scale dense GAN model to enhance underwater images. Li et al. [25] collected an underwater image dataset with its subjectively better chosen restored results and proposed a gated fusion network to improve underwater images enhancement. A physical model for simulating real underwater images according to different water body types is proposed by Li et al. [26]. In total, they trained ten models together for underwater image enhancement, named UWCNN, each of which corresponds to one group of synthesized underwater images. Collected underwater images usually determine the performance of these underwater image enhancement models for training. Thus, it is challenging to have enough underwater images to represent real-world conditions. For this reason, the generalization capability and robustness of current deep learning-based models for underwater image enhancement/restoration are limited and somewhat dissatisfied.

Unlike the previously mentioned methods, this work proposes a novel idea that combines the conventional HE [4] with a PDM-based restoration method to strike a balance between “whether the image has good contrast” and “whether the image color is recovered” through convex optimization techniques. The proposed method using the HE approximation based on the PDM considers the physics characteristics of nature images to make the color restoration consistent among pixels while keeping the distributions of image intensities uniform across the entire IL range for better contrast.

The rest of the paper is organized as follows. Section 2 describes related works. The HE and how to use PDM to restore underwater images are briefly reviewed in Section 3. The proposed method is described in Section 4. The experimental results are reported in Section 5. Finally, Section 6 summarizes the conclusions.

## 2. Related Work

### 2.1. Histogram Equalization

Due to the absorption, scattering, and attenuation of propagated light with distance from the camera, the contrast of underwater imaging will be significantly reduced and cause fogging. CE is a helpful method to improve the visual quality of such images and bring out image details that would be otherwise invisible. In the field of CE, there are many related extension techniques, and global HE [4] is one of the most widely used because of its simplicity of implementation and generally satisfactory results. Global HE attempts to adjust the intensity levels (IL) presented in the image to be well-distributed over the IL range using a transformation function derived from the image cumulative distribution function (CDF). However, it can lead to under-enhancement or over-enhancement of the image. Significant peaks in the histogram may occupy wider ranges in the mapping transformation, leaving a relatively narrow range for other intensity levels. In addition, using global HE may also increase the noise in the input images and create unwanted artifacts. Much research has been conducted to improve global HE. In 1977, Kim [27] proposed brightness preserving bi-histogram equalization (BBHE) to maintain the original brightness in CE. It splits the image histogram into two sub-histograms based on the mean intensity of the global HE. The idea of mean brightness preservation, such as Dualistic sub-image histogram equalization (DSIHE) [28] which is similar to BBHE, separates the histogram according to the median intensity. Then, multiple histogram separations such as multi-peak histogram equalization (MPHE) [29], recursive mean separation histogram equalization (RMSHE) [30], and recursive sub-image histogram equalization (RSIHE) [31] are sequentially performed. They are proposed to maintain mean brightness.

In keeping mean brightness while avoiding added noise in global HE, the image histogram can be modified by cropping to control the CE rate. The method of HE with bin underflow and bin overflow (BUBOHE) [32] was presented to regulate the gradient of the mapping function by setting the lower and upper bounds of the calculated probability density function (PDF). Besides, gain-controllable clipped histogram equalization (GC-CHE) [33] is a clipped HE method using gain control. Furthermore, a hybrid histogram modifications (HM) method, bi-histogram equalization with a plateau limit (BHEPL) [34], was presented to integrate the histogram separation and clipping techniques for CE. Adaptive gamma correction with weighting distribution (AGCWD) [35] is another hybrid HM method for CE that combines the gamma correction [36] and global HE to keep the original mean brightness while curbing the under-enhancement and over-enhancement.

In subsequent studies, a 2D HE method that exploits the contextual information around each pixel in the input image was proposed and applied to global HE instead of using the 1D intensity histogram. In 2011 and 2012, Celik proposed two-dimensional histogram equalization (2DHE) [37] and contextual and variational contrast enhancement (CVCE) [38], respectively. Both use a sliding window to compute a two-dimensional histogram that counts the neighbor pixel intensities for each intensity level to enhance the contextual features further.

The global HE described above may effectively enhance contrast for most natural images. However, it may not be enough for restoring underwater images because they do not only suffer low contrast but color distortions.

### 2.2. Pdm-Based Restoration Methods

Many underwater image restoration methods have been proposed in recent years. Since our approach is relevant to PDM-based restoration, we introduce these relevant methods here. Since underwater images resemble hazy images in some cases, researchers have employed dehazing methods to deal with underwater image restoration. For instance, single image dehazing based on the dark channel prior [8] assumes that dark pixels exist in most local patches of natural images without haze. Based on this assumption, one can estimate the haze transmission and ambient light to restore hazy images based on the PDM. The underwater DCP (UDCP) [7] method observed that the absorption rate of red light is larger and modified DCP to fit underwater scenes. Similarly, the red channel method [14] is considered a dark channel method to recover degraded images by restoring the contrast and colors associated with short wavelengths. Peng et al. proposed a depth estimation method for underwater scenes based on image blurriness and light absorption (IBLA) [10], restoring underwater images. Besides, Peng et al. also proposed a Generalized Dark Channel Prior (GDCP) [9] approach for image restoration, which incorporates adaptive color correction into PDM. This approach can be reduced to several DCP variants for different special situations of turbid medium and ambient lighting conditions. Importantly, underwater images usually present blur and color casts caused by light absorption and scattering in a water medium. Therefore, the underwater light attenuation prior method (ULAP) [39] applys the correct depth map, the transmission maps (TMs), and the background light (BL) for RGB lights to recover the true scene radiance under the water through an image formation model. Song et al. [40] restores underwater images using transmission map optimization. However, these image restoration methods do not consider further enhancing the contrast of the restored images, sometimes presenting unsatisfying restored results.

## 3. Background Reviews

### 3.1. Histogram Equalization

Let *I* be the input image and $I\left(x\right)\in \left[0,L-1\right]$ be the intensity of the input image at pixel *x*, where $L=2^{b}$ for an image with a *b*-bit IL. Let the image histogram $H=\left[h_{0},h_{1},\ldots ,h_{L-1}\right]\in N^{L}$. The probability density function $P={\left[p_{0},p_{1},\ldots ,p_{L-1}\right]}^{T}\in R_{+}^{n}$ of *I* can be depicted in a vector form as:(1)$$P=\frac{H}{1^{T}H},$$
where $1={\left[1,1\ldots 1\right]}^{T}\in N^{L}$
The CDF $C={\left[c_{0},c_{1},\ldots ,c_{L-1}\right]}^{T}\in R_{+}^{n}$ of the image is computed as $c_{l}=\sum_{i=0}^{l}p_{i}$. In HE, the transform function $T_{f}$ is derived as:(2)$$T_{f}\left(l\right)=\left\lfloor \left(L-1\right)c_{l}+0.5\right\rfloor ,$$
where l∈[0,L−1]. Finally, each distinct IL *l* of the input image *I* is remapped to a corresponding output IL via the transform function in Equation (Equation 2).

### 3.2. Pdm-Based Image Restoration

A simple PDM-based method for underwater images is given as [8,9]:(3)$$I^{c}\left(x\right)=J^{c}\left(x\right)t\left(x\right)+B^{c}\left(1-t\left(x\right)\right),$$
where $I^{c}$ is the observed intensity in the channel *c*, with *c* being one of the R, G, B channels, $J^{c}$ is the scene radiance, $B^{c}$ is the background light (BL), and *t* is the transmission map. The transmission describes how much light is not attenuated from traveling through the medium and reaches the camera. Besides, it can be formulated as an exponential decay term of the scene depth:(4)$$t\left(x\right)=e^{-\beta d(x)},$$
where β is the attenuation coefficient, and d(x) represents the corresponding depth for the pixel at *x*. Thus, if a scene point is farther from the camera its *t* gets smaller.

Let $S_{0.1}$ be the set of the positions of the top 10% brightest pixels in $I_{dark}$, where $I_{dark}\left(x\right)=\min_{y\in \Omega (x)}\left\{\min_{c}I^{c}\left(y\right)\right\}$. The estimated BL $B^{c}$ can be calculated by:(5)$$B^{c}=\frac{1}{\left|S_{0.1}\right|}\sum_{x\in S_{0.1}}I^{c}\left(x\right),\;\;\;\;\;\;c\in \left\{r,g,b\right\}.$$

Based on the DCP, the depth map t(x) is estimated [8] as:(6)$$t\left(x\right)=1-\min_{y\in \Omega (x)}\left\{\min_{c\in \{R,G,B\}}\frac{I^{c}\left(y\right)}{B^{c}}\right\},$$
where Ω(x) is a local patch centered at *x*. In our work, a 15×15 patch is adopted. Since the depth map has blocking artifacts, it can be refined by median filtering. Finally, plugging $I^{c}$, *t* and $B^{c}$ into Equation (Equation 3), the scene radiance $J^{c}$ can be recovered as:(7)$$J^{c}\left(x\right)=\frac{I^{c}\left(x\right)-B^{c}}{\max\left(t\left(x\right),r_{0}\right)}+B^{c},\;\;\;\;\;\;c\in \left\{r,g,b\right\},$$
where $r_{0}$ is empirically set from 0.1 to increase the exposure of *J* for display.

Figure 1 shows two sets of underwater images: underwater images and their enhanced images obtained using the HE and the PDM-based method [9] respectively. We can observe that applying the HE to the images produces better contrast; however, the colors of these enhanced images are distorted. By contrast, the PDM-based method restores degraded images by reversing the image formation process, but the restored images may have low contrast. Therefore, the proposed method aims at combining the HE and PDM to obtain restored results with good contrast.

## 4. Proposed Method

In this section, we discuss the details of the proposed approach. The overall framework of the proposed method is depicted as Figure 2, where it combines HE and PDM-based restoration to balance between the input image’s HE transformation function and PDM mapping distribution via convex optimization. To this end, it outputs the enhanced image using the final transformation function generated.

To combine the HE and PDM, we first consider the HE part. Since the transformation function $T_{f}$ is one-to-one, it can be represented in a vector form $T={\left[t_{0},t_{1},...,t_{L-1}\right]}^{T}\in N^{L}$ as:(8)$$\begin{array}{cc} T & =\lfloor \left(L-1\right){\left[c_{0},c_{1},\ldots ,c_{L-1}\right]}^{T}+0.5\rfloor \\ \; & =\lfloor \left(L-1\right)[p_{0},p_{0}+p_{1},\ldots ,\sum_{i=0}^{L-1}p_{i}{]}^{T}+0.5\rfloor , \end{array}$$
which is referred to as the transformation matrix. According to Equation (Equation 8), the difference of two consecutive ILs can be given by:(9)$$\begin{array}{cc} t_{k}-t_{k-1} & =\left\lfloor \left(L-1\right)c_{k}+0.5\right\rfloor -\left\lfloor \left(L-1\right)c_{k-1}+0.5\right\rfloor \\ \; & \approx \lfloor \left(L-1\right)p_{k}\rfloor . \end{array}$$

Then, the transformation function *T* can be further denoted as:(10)$$\begin{array}{c} RT=\left[\begin{array}{ccccc} 1 & 0 & \cdots & 0 & 0\\ -1 & 1 & \cdots & 0 & 0\\ \vdots & \vdots & \ddots & \vdots & \vdots \\ 0 & 0 & \cdots & 1 & 0\\ 0 & 0 & \cdots & -1 & 1 \end{array}\right]T\approx \left\lfloor \left(L-1\right)P\right\rfloor =\hat{P}, \end{array}$$
where $R\in Z^{L\times L}$ is a tridiagonal matrix as shown above.

For the PDM part, since the recovered image is derived by Equation (Equation 7), there is no way to find a one-to-one function to describe it because each distinct IL *l* of the input image $I^{c}$ may be mapped to multiple ILs in the recovered scene radiance *J*. Hence, in order to combine the PDM with the HE, it can be assumed that the transformation matrix $T^{c}=\left[t_{0}^{c},t_{1}^{c},...,t_{L-1}^{c}\right]$ is an independent Gaussian random vector, where $T^{c}∼N\left(\bar{T^{c}},\Sigma_{T^{c}}\right)$, and $\bar{T^{c}}={\left[\bar{t_{0}^{c}},\bar{t_{1}^{c}},\ldots ,\bar{t_{L-1}^{c}}\right]}^{T}\in R^{L}$ and $\Sigma_{T^{c}}=diag\left\{\sigma_{t_{k}^{c}}^{2}\right\}\in R^{L\times L}$, ∀k∈[0,L−1], where $\bar{t_{k}^{c}}$ and $\sigma_{t_{k}^{c}}^{2}$ represent the mean and the variance of the random variable $t_{k}^{c}$, respectively.

However, not every element in the $\bar{T^{c}}$ and $\Sigma_{T^{c}}$ is valid because there may be empty bins in the histogram of the input image, i.e., the elements in $T^{c}$ that corresponds to empty bins are regarded as invalid. Thus, a matrix $E\in R^{\tilde{n}\times L}$ is generated to get rid of those elements in $T^{c}$ that do not have valid Gaussian distributions, where $\tilde{n}$ is the number of distinct ILs in the input image *I*. Assume each element $t_{k}^{c}$ in $T^{c}$ corresponds to a row vector $e_{k}^{c}\in R^{1\times L}$ where only the ${\left(k+1\right)}_{th}$ element is 1 and the rest are 0. E is given by:(11)$$\begin{array}{c} E=\left[\begin{array}{c} \vdots \\ e_{k}^{c}\\ \vdots \end{array}\right]\in R^{\tilde{n}\times L}, \end{array}$$
where $t_{k}^{c}$ is valid, and k∈[0,L−1]. Hence, the assumption is modified as $ET^{c}∼N\left(E\bar{T^{c}},E\Sigma_{T^{c}}E^{T}\right)$.

To this end, the objective function $f_{0}\left(T^{c}\right)$ of combining the HE and the PDM is formulated as:(12)$$\begin{array}{c} \begin{array}{cc} \; & \mathrm{minimize}\;∥RT^{c}-\hat{P}{∥}_{2}-\alpha \log G\left(ET^{c}\right)\\ \; & \mathrm{subject}\mathrm{to}\;RT^{c}⪰0,\\ \; & \;\;\;t_{0}^{c}=0,\\ \; & \;\;\;t_{L-1}^{c}=L-1,\;c\in \left\{r,g,b\right\}, \end{array} \end{array}$$
where α≥0 is a control parameter, and the multivariate Gaussian distribution *G* is given by:(13)$$G\left(ET^{c}\right)=\frac{1}{\sqrt{{\left(2\pi \right)}^{\tilde{n}}\left|\tilde{\Sigma_{T^{c}}}\right|}}\exp\left\{-\frac{∥{\left(\tilde{\Sigma_{T^{c}}}\right)}^{-\frac{1}{2}}E\left(T^{c}-\bar{T^{c}}\right){∥}_{2}^{2}}{2}\right\},$$
where $\tilde{\Sigma_{T^{c}}}=E\Sigma_{T^{c}}E^{T}$.

The objective function $f_{0}\left(T^{c}\right)$ has two terms: the HE term $∥RT^{c}-\hat{P}{∥}_{2}$ and the PDM-based Gaussian distribution term $\log G(ET^{c})$. Since *G* is log-concave, and the logarithm is monotonically increasing, logG is adopted to regularize the HE term by penalizing the transformation matrix $T^{c}$ being different from the PDM-based distribution and to make the objective function $f_{0}$ convex as well.

To solve Equation (Equation 12), the PDM term can be simplified as:(14)$$\begin{array}{cc} \alpha & \log G(ET^{c})\\ = & \frac{\alpha }{2}\left[\tilde{n}\log2\pi +\log|\tilde{\Sigma_{T^{c}}}|+∥{\left(\tilde{\Sigma_{T^{c}}}\right)}^{-\frac{1}{2}}E\left(T^{c}-\bar{T^{c}}\right){∥}_{2}^{2}\right]\\ = & \tilde{\alpha }\left[∥{\left(\tilde{\Sigma_{T^{c}}}\right)}^{-\frac{1}{2}}E\left(T^{c}-\bar{T^{c}}\right){∥}_{2}^{2}+C\right], \end{array}$$
where $\tilde{\alpha }=\frac{\alpha }{2}$, and $C=\tilde{n}\log2\pi +\log\left|\tilde{\Sigma_{T^{c}}}\right|$, which is constant in $f_{0}$ and thus can be dropped. Therefore, by introducing two slack variables, Equation (Equation 12) is equivalent to:(15)$$\begin{array}{c} \begin{array}{cc} \; & \mathrm{minimize}\;t+\tilde{\alpha }g\\ \; & \mathrm{subject}\mathrm{to}\;RT^{c}⪰0,\\ \; & \;\;\;t\geq ∥RT^{c}-\hat{P}{∥}_{2},\\ \; & \;\;\;g\geq ∥{\left(\tilde{\Sigma_{T^{c}}}\right)}^{-\frac{1}{2}}E\left(T^{c}-\bar{T^{c}}\right){∥}_{2}^{2},\\ \; & \;\;\;t_{0}^{c}=0,\\ \; & \;\;\;t_{L-1}^{c}=L-1,\;c\in \left\{r,g,b\right\}, \end{array} \end{array}$$
which can be solved by using CVX, a package for specifying and solving convex programs [41,42]. Through minimizing $f_{0}\left(T^{c}\right)$ with an appropriate $\tilde{\alpha }$, ranging from $10^{-4}$ to $10^{-5}$ in this paper, the HE approximation based on the PDM can further improve the enhanced image quality.

## 5. Experimental Results

In this section, we first describe the implementation details, then compare the proposed method against the HE and the PDM-based methods in an image enhancement context.

### 5.1. Experiment Settings

In this work, we employ the UIEB [25] dataset to test our method. The UIEB dataset is a real-world underwater image dataset consisting of 890 underwater images with the corresponding high-quality reference images, including haze-like, greenish, and blueish underwater images. We compare our proposed method against 7 state-of-the-art underwater image restoration methods, including the Fusion-based [43], Retinex-based [44], IBLA [10], AGC [45], GDCP [9], BL-TM [40], and UWCNN [26]. We set $\tilde{\alpha }=5\times 10^{-5}$ and $r_{0}$ = 0.1 in our method for all the experiments. For all the other compared methods, we follow their default settings to process underwater images.

The performance of the compared methods is evaluated qualitatively and quantitatively. We choose three prominent cases for the qualitative assessment, comprising haze-like, greenish, and blueish underwater images. For the quantitative evaluation, we adopt two no-reference quality metrics for accessing the quality of underwater images: Underwater Color Image-Quality Evaluation (UCIQE) [46] and Natural Image Quality Evaluator (NIQE) [47]. The UCIQE linearly combines the variation of chrominance, average saturation, and luminance contrast to measure underwater image quality. Higher UCIQE values indicate better visual perceptual quality. The NIQE measures deviations of the tested image from regularities of natural scene statistics. Lower NIQE values suggest better image quality.

### 5.2. Qualitative Assessment

We chose representative cases from the UIEB dataset to demonstrate visual comparison enhancement results, including haze-like, greenish, and blueish underwater images. The competing methods either do not restore color distortions or produce low-contrast images. By contrast, the results generated by our method look natural with better contrast.

Figure 3 demonstrate restoration results for a common haze-like blueish underwater image. Fusion-based [43], Retinex-based [44], AGC [45], and UWCNN [26] successfully dehaze (dim) the image to increase its contrast. However, the blueish color cast remains. IBLA [10], in contrast, brighten the image and restore its color a little. BL-TM [40] brightens the image more, thus making it even hazier. GDCP [9] performs poorly here since it over-enhances the brightness of the image, making it unnatural. The proposed method restores the image’s color and enhances its contrast, presenting a visually satisfying result.

Figure 4 shows enhanced results for another blueish underwater image, where most methods work fine. Out of the compared methods, UWCNN [26] seems to do little for the restoration to the image. GDCP [9] and BL-TM [40] distort the color of the restored images. Fusion-based [43], Retinex-based [44], IBLA [10], AGC [45], and the proposed method all perform well for this case.

Figure 5 shows the comparisons of enhanced results on a haze-like blueish underwater image. As can be seen, the methods [43,44,45] enhance the image’s contrast but do not work well on color restoration. UWCNN [26] makes the output darker but seems not to work at all. IBLA [10] and GDCP [9] improve the contrast but introduce color distortions. BL-TM [40] corrects the color a bit but fails to recover its contrast. The proposed method performs better regarding contrast and color balance together.

Figure 6 and Figure 7 shows the comparisons of processed results on a haze-like greenish underwater image, where Fusion-based [43] and Retinex-based [44] methods remove the greenish color cast but the restored images look a little washed out and unnatural. IBLA [10], AGC [45], and UWCNN [26] fail to the restore color of the image. GDCP [9] overcorrects the input image with more red color added. BL-TM [40] again corrects the color but does not perform well on contrast. The proposed method works better than the other methods visually.

As can be seen from the above results, most methods can work in terms of color restoration or contrast enhancement to some extent. The proposed method can effectively remove the haze and possible color casts and improve the contrast without over-saturation and over-enhancement. Notably, we also show the objective performance here on each restored image, where our results achieve the best UCIQE/NIQE scores.

### 5.3. Quantitative Assessment

Table 1 displays objective comparisons among the methods on the UIEB dataset [25] based on UCIQE [46] and NIQE [47], where the proposed method performs favorably against the other state-of-the-art approaches.

To sum up, we can see by solving the convex problem formulated in Equation (Equation 15) with an appropriate control parameter $\tilde{\alpha }$, the proposed method can effectively combine HE and PDM to present satisfying enhancement results.

### 5.4. Runtime Assessment

We compared the average runtime for all the compared methods on the UIEB dataset [25], shown in Table 2. As can be seen, the AGC [45] and Retinex [44] runs faster than the other methods. IBLA [10] is the slowest. Our proposed method takes 4.8 s on average.

### 5.5. Application to Feature Matching

Figure 8 shows an example of feature-point matching. We match an underwater image with its rotated version using SIFT [48] to demonstrate how an image enhancement helps improve the image’s quality and increase the matching feature points. In Table 3, we can see the number of feature matching points obtained using different enhancement methods, where our method presents more key feature points.

## 6. Conclusions

We proposed to enhance underwater images using a transformation function derived by a convex combination of Histogram Equalization and Physics-based Dichromatic modeling. The proposed method can generate visually pleasing results with restored color and better contrast. The proposed method can outperform state-of-the-art underwater image restoration approaches based on the qualitative and quantitative experimental results.

## Figures and Tables

**Figure 1 sensors-22-02168-f001:**
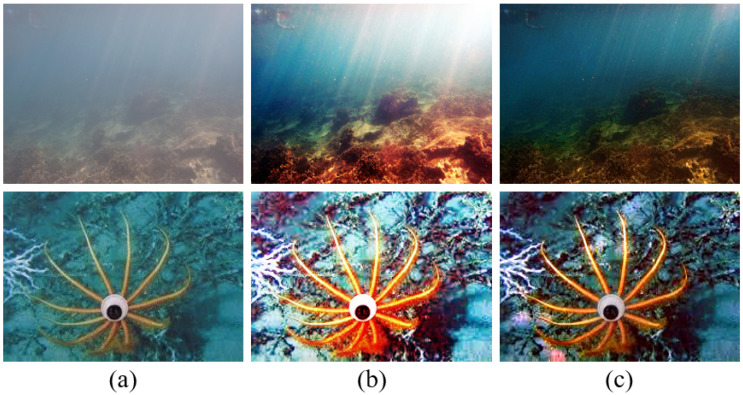
A comparison of underwater image enhanced results. (**a**) Underwater images. The enhanced images obtained using (**b**) the HE [4], and (**c**) ref. [9] for the underwater image.

**Figure 2 sensors-22-02168-f002:**
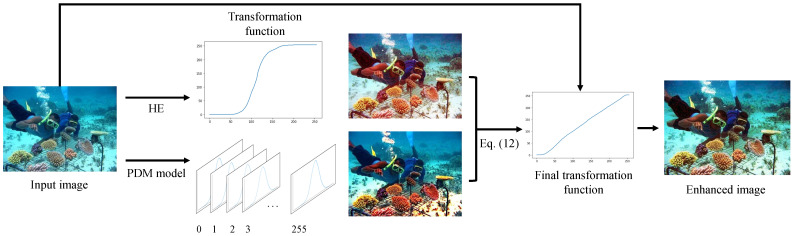
The overall framework of the proposed method.

**Figure 3 sensors-22-02168-f003:**
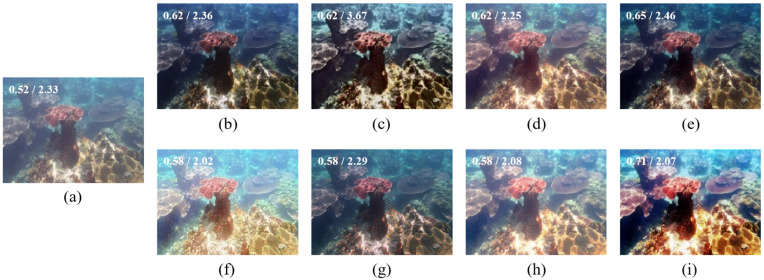
(**a**) Dark bluish underwater image, and its restored/enhanced results obtained using (**b**) Fusion-based [43], (**c**) Retinex-based [44], (**d**) IBLA [10], (**e**) AGC [45], (**f**) GDCP [9], (**g**) UWCNN [26], (**h**) BL-TM [40], and (**i**) the proposed method. The number on the top-left corner of each image refers to its UCIQE/NIQE score.

**Figure 4 sensors-22-02168-f004:**
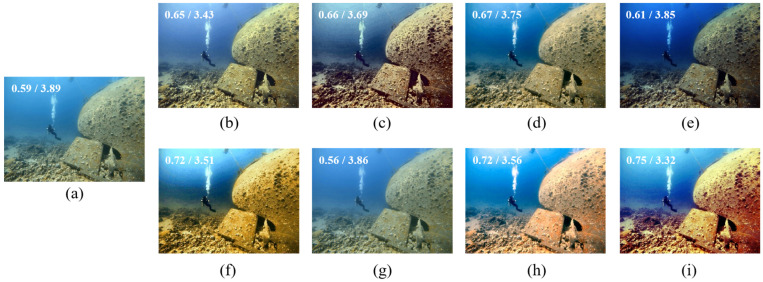
(**a**) Bright bluish underwater image, and its restored/enhanced results obtained using (**b**) Fusion-based [43], (**c**) Retinex-based [44], (**d**) IBLA [10], (**e**) AGC [45], (**f**) GDCP [9], (**g**) UWCNN [26], (**h**) BL-TM [40], and (**i**) the proposed method. The number on the top-left corner of each image refers to its UCIQE/NIQE score.

**Figure 5 sensors-22-02168-f005:**
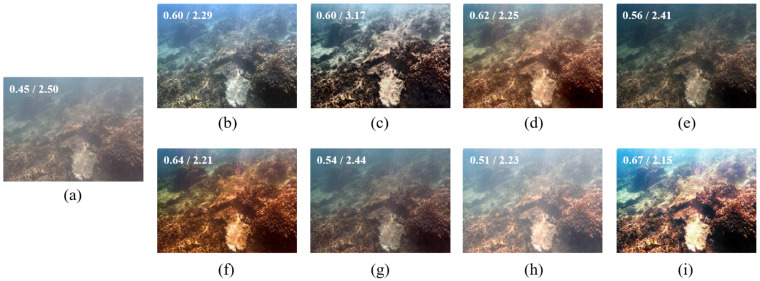
(**a**) Hazy and bluish underwater image, and its restored/enhanced results obtained using (**b**) Fusion-based [43], (**c**) Retinex-based [44], (**d**) IBLA [10], (**e**) AGC [45], (**f**) GDCP [9], (**g**) UWCNN [26], (**h**) BL-TM [40], and (**i**) the proposed method. The number on the top-left corner of each image refers to its UCIQE/NIQE score.

**Figure 6 sensors-22-02168-f006:**
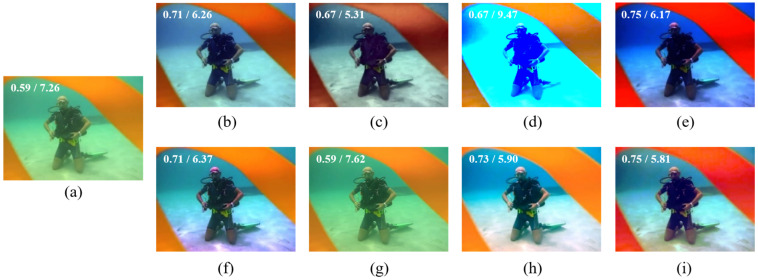
(**a**) Greenish underwater image with a diver and ribbons, and its restored/enhanced results obtained using (**b**) Fusion-based [43], (**c**) Retinex-based [44], (**d**) IBLA [10], (**e**) AGC [45], (**f**) GDCP [9], (**g**) UWCNN [26], (**h**) BL-TM [40], and (**i**) the proposed method. The number on the top-left corner of each image refers to its UCIQE/NIQE score.

**Figure 7 sensors-22-02168-f007:**
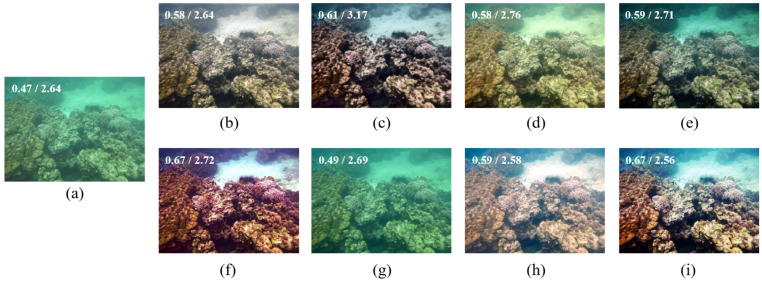
(**a**) Greenish underwater image, and its restored/enhanced results obtained using (**b**) Fusion-based [43], (**c**) Retinex-based [44], (**d**) IBLA [10], (**e**) AGC [45], (**f**) GDCP [9], (**g**) UWCNN [26], (**h**) BL-TM [40], and (**i**) the proposed method. The number on the top-left corner of each image refers to its UCIQE/NIQE score.

**Figure 8 sensors-22-02168-f008:**
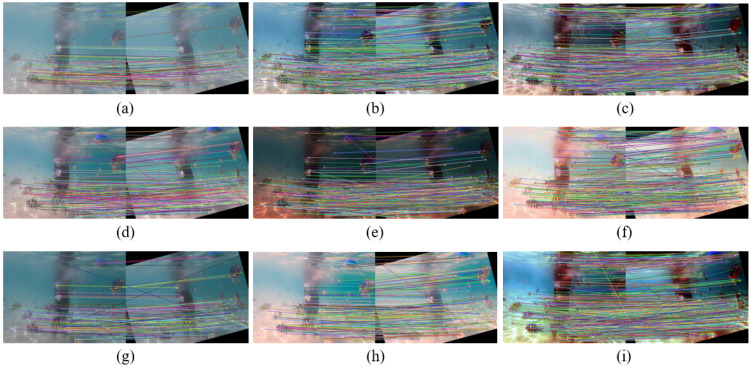
An example of feature-point matching. We match an underwater image with its rotated version using SIFT [48] to demonstrate how an image enhancement helps improve the image’s quality and increase the matching feature points. (**a**) Original underwater image pair, and its enhanced version using (**b**) Fusion-based [43], (**c**) Retinex-based [44], (**d**) IBLA [10], (**e**) AGC [45], (**f**) GDCP [9], (**g**) UWCNN [26], (**h**) BL-TM [40], and (**i**) the proposed method.

**Table 1 sensors-22-02168-t001:** The UCIQE [46] and NIQE [47] scores for all the compared methods on the UIEB dataset [25]. The best scores are in bold.

Methods	UCIQE ↑	NIQE ↓
input	0.52	4.31
Fusion-based [43]	0.59	3.85
Retinex-based [44]	0.60	4.09
IBLA [10]	0.60	3.90
AGC [45]	0.62	4.04
GDCP [9]	0.60	3.88
UWCNN [26]	0.52	4.20
BL-TM [40]	0.62	4.00
Proposed	**0.64**	**3.79**

**Table 2 sensors-22-02168-t002:** Average runtime in seconds on the UIEB dataset [25].

	Fusion	Retinex	IBLA	AGC	GDCP	UWCNN	BL-TM	Proposed
Runtime ↓	5.58	0.66	41.74	0.65	1.00	9.25	22.17	4.80

**Table 3 sensors-22-02168-t003:** Numbers of matching feature points in Figure 8.

	Input	Fusion	Retinex	IBLA	AGC	GDCP	UWCNN	BL-TM	Proposed
#Feature pts ↑	134	747	923	473	303	591	180	290	1005

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
