# Peer review of "Underwater Image Enhancement Based on Histogram-Equalization Approximation Using Physics-Based Dichromatic Modeling"

_sensors, 2022, doi:10.3390/s22062168_

Round 1
Reviewer 1 Report
This manuscript presents a relatively effective underwater image enhancement method that combines histogram-equalization approximation and physics-based dichromatic modeling. The innovative approach not only enhances image contrast but also restores distorted color. Before accepting this manuscript, I suggest revising it from the following aspects.
- In the introduction (Section 1), the author describes the shortcomings of the existing deep learning methods well, but the description of the traditional methods is slightly insufficient. Only the shortcomings of previous related methods based on histogram-equalization (HE) and dark channel prior (DCP) are introduced. I suggest that their brief introduction could be placed in the second section and add some state-of-the-art physics-based underwater image enhancement/restoration methods in the first section (Introduction).
- A general framework diagram to illustrate the implementation of the method is missing in this manuscript. In order to illustrate the innovation of this paper more vividly and concretely, I suggest that the author add it.
- In the experimental results (Section 5), the authors have shown a comparison of eight methods on no-reference evaluation metrics (UCIQE and NIQE). However, there is a lack of full-reference quantitative evaluation. I suggest that the authors can add some relevant metrics, such as PCQI, PSNR, and SSIM.
- The purpose of underwater image enhancement is suitable to be combined with other computer vision applications as an effective pre-processing. Therefore, I suggest adding experiments on image stitching to reconstruct wide-field underwater scenes, which can more convincingly demonstrate that underwater images enhanced by this method are more suitable for subsequent underwater research. Specifically, the reconstruction process can refer to the paper entitled " Enhancement-Registration-Homogenization (ERH): A Comprehensive Underwater Visual Reconstruction Paradigm", which has been published on IEEE Transactions on Pattern Analysis and Machine Intelligence and the source code is available. It is reproducible.
- For reader readability and good visual structure, I recommend that the author rearrange the layout of images and tables. For example, I hope to see this Reference and Conclusion at the end, instead of seeing various experimental results.

Reviewer 2 Report
The paper describes a new method for the enhancement of underwater images, using both histogram equalization and physics-based dichromatic modeling.
The experiments performed on a classical image database show that the proposed method offers better non-reference image quality metrics evaluation results than state of the art methods.
One would appreciate the evaluation of the newer, SQUID dataset (http://csms.haifa.ac.il/profiles/tTreibitz/datasets/ambient_forwardlooking/index.html) especially in terms of coloc consistency between image pairs.
In order to really understand the full performance of the method one would require also a computational complexity evaluation of the proposed method as compared to state of the art.
Also, the paper would benefit from a supplemental carefull proofreading.
Reviewer 3 Report
- Please revise the list sentence in the abstract section. The method was developed or not?
- Paragraph on line 60. Please highlight the differences between the propose method and state-of-the-art. Clearly highlight the novelty of this work.
- Please revise Figure 1 caption.
- Section 5.1, please clarify how these 7 state-of-the-art methods were applied to the selected set of images? Did you have access to the code? Did the paper's authors provide the images? Did the authors implement these methods?
- What was used to compute the UCIQE and NIQE metrics? Again code provide by the authors or the metrics were implemented? It is important that all authors use the same implementation.
- Besides Table 1, please insert two image with the results for every image in the dataset and for each of the 8 methods mentioned in Table 1, for the two metrics. I.e., the data used to generate Table 1.
Round 2
Reviewer 1 Report
The authors have made revisions as suggested by me, and I agree to recommend acceptance of this manuscript.